

# PPP-based Swarm kinematic orbit determination

Le Ren[1] and Steffen Schön[1]

[1]Leibniz Universität Hannover, Institut für Erdmessung (IfE), Schneiderberg 50, 30167 Hannover, Germany

**Correspondence:** Le Ren (ren@ife.uni-hannover.de)

**Abstract.** ESA's Swarm mission offers excellent opportunities to study the ionosphere and to bridge the gap in gravity field recovery between GRACE and GRACE-FO. In order to contribute to these studies, at IfE Hannover, a software based on precise point positioning (PPP) batch least-squares adjustment is developed for kinematic orbit determination. In this paper, the main achievements are presented.

The approach for the detection and repair of cycle slips caused by ionospheric scintillation is introduced, which is based on the Melbourne-Wübbena and ionosphere-free linear combination. The results show that around 95% cycle slips can be repaired and the majority of the cycle slips occur on $L_2$. After the analysis and careful preprocessing of the observations, one year kinematic orbits of Swarm satellites from September 2015 to August 2016 are computed with the PPP approach. The kinematic orbits are validated with the reduced-dynamic orbits published by ESA in Swarm Level 2 products and SLR

measurements. The differences between IfE kinematic orbits and ESA reduced-dynamic orbits are at the 1.5 cm, 1.5 cm and 2.5 cm level in the along-track, cross-track and radial directions, respectively. Remaining systematics are characterised by spectral analyses, showing once per revolution period. The external validation with SLR measurements shows RMS errors at the 4 cm level. Finally, fully populated covariance matrices of the kinematic orbits obtained from 30 s, 10 s and 1 s data rate are discussed. It is shown that for data rates larger than 10 s, the correlation should be taken into account.

*Copyright statement.* TEXT

# 1  Introduction

The Swarm mission was launched on November 22, 2013 and is ESA's first constellation of satellites to study the dynamics of the Earth's magnetic field and its interaction with the Earth system (Friis-Christensen et al. , 2008). This mission consists of three identical satellites in near-polar low Earth orbits (LEO). The two satellites Swarm A and C fly almost side-by-side

at an initial altitude of 460 km, the third Swarm satellite B flies in a higher orbit of about 530 km. All the three satellites are equipped with a set of six core instruments: absolute scalar magnetometer (ASM), vector field magnetometer (VFM), electric field instrument (EFI), star tracker (STR), accelerometer (ACC) and GPS receiver (GPSR). In order to take full advantage of the data information provided by this constellation, Precise Orbit Determination (POD) is necessary, which is obtained with data from the high precision 8-channel dual-frequency GPS receiver. In addition, each satellite has a laser retro-reflector,





which makes the independent validation of the GPS-derived orbits with Satellite Laser Ranging (SLR) possible. In recent years, kinematic orbit determination has attracted much attention (Švehla and Rothacher , 2003), (Zehentner and Mayer-Gürr , 2015), (Jäggi et al. , 2016). Compared to the reduced-dynamic approach, the kinematic approach is a purely geometrical approach without using any information on LEO satellite dynamics (e.g. gravity field, air-drag, etc.). Therefore, kinematic orbits are

preferred for the gravity field determination.

A first performance evaluation of the RUAG 8-channel GPS receiver was carried out by Zangerl et al. (2014), showing unexpected results and loss of lock due to ionospheric scintillations. Buchert et al. (2015) and Xiong et al. (2016) empirically linked the loss of GPS signal to ionospheric plasma irregularities. Results from more than one year of data using reduced-dynamic and kinematic approaches were reported in van den IJssel et al. (2015) and Jäggi et al. (2016). A comparison with

SLR allows an assessment of the orbit accuracy of about 2.5 cm rms for ESA reduced-dynamic orbits and 4 cm for ESA kinematic orbits. The 3D differences between the reduced-dynamic and kinematic orbits are at the 4-5 cm level, with larger differences close to the geomagnetic poles and along the geomagnetic equator (van den IJssel et al. , 2015). An azimuth and elevation dependent PCV map is generated during flight using the residual approach (van den IJssel et al. , 2015), (Jäggi et al. , 2016). In order to collect more observations and improve the robustness under ionospheric scintillation, the receiver settings

have been adjusted several times during the Swarm mission. The impact of the first update on POD performance is analysed in van den IJssel et al. (2016), showing that a lower GPS elevation cut-off angle and wider bandwidths of the tracking loops improve the performance for both orbit types. Gravity field recovery benefits also from the update (Dahle et al. , 2017).

In this contribution, the developments for kinematic POD of Swarm satellites at IfE are reported. This approach is based on the Precise Point Positioning (PPP) technique (Zumberge et al. , 1997). This differs from the raw measurements approach by

Institute of Geodesy (IfG) of TU Graz (Zehentner and Mayer-Gürr , 2015), phase-only approach by Astronomical Institute of University Bern (AIUB) (Jäggi et al. , 2016) or Bayesian weighted least-squares estimator of ESA (van den IJssel et al. , 2015).

Since the GPS data quality is decisive for the obtainable orbit accuracy, we first analyse in Section 2 the in-flight performance of the Swarm on-board GPS receivers, e.g. the tracking performance and observation noise, especially under the influence of ionospheric scintillation. A mandatory step in the preprocessing is detecting and, if possible, repairing the cycle slips in carrier

phase observations in order to reduce the number of ambiguities and therefore strengthen the orbit. Due to the large phase noise caused by ionospheric scintillations, cycle slips in Swarm receivers are difficult to detect at ionospheric-active areas. A approach based on the Melbourne-Wübbena and ionosphere-free linear combination is introduced for this task. In Section 3, we introduce the adopted method and models for precise kinematic orbit determination by PPP . The obtained orbit as well as its fully populated covariance matrix are presented. Unlike CHAllenging Minisatellite Payload (CHAMP) or Gravity Recovery

and Climate Experiment (GRACE), the Swarm GPS receiver provides observations every second since 15 July 2014, which gives us the opportunity to study the temporal correlation of the coordinates with 1 s real observations. The kinematic orbits are compared with the ESA reduced-dynamic orbits and SLR measurements. Thanks to a more strict data screening strategy and cycle slip detection, our kinematic orbits show a smaller RMS error compared to ESA kinematic orbits. Section 4 gives a short conclusion of this paper.



## 2 Swarm GPS data qualities

Being a purely geometrical approach without using any dynamic models on LEO satellites, the quality of the kinematic orbits relies completely on the GPS observations and it is sensitive to measurement errors and the observation geometry (Švehla and Rothacher , 2003). High quality data is the prerequisite for precise kinematic orbits. Therefore, in this section we first analyse

the Swarm GPS receiver data quality and introduce our approach for cycle slip detection/repair and outlier screening.

### 2.1 Tracking performance

The Swarm on-board dual-frequency GPS receivers are developed by RUAG Space (Zangerl et al. , 2014). They have 8 channels for tracking GPS satellites, which is less than GRACE and Gravity field and steady-state Ocean Circulation Explorer (GOCE) receivers, with 10 and 12 channels, respectively. In order to collect more observations, the field of view was increased step by

step from $10°$ cut-off angle at the beginning of the mission to $2°$ cut-off angle since 06 May 2015 for all three Swarm satellites (van den IJssel et al. , 2015). Several updates on the bandwidth of the tracking loop were applied to improve the robustness of the tracking in difficult environmental situations. The fast reacquisition of the $L_1$-carrier tracking was enabled by raising the retry counter from zero to five (ESA , 2015). These updates are shortly summarized in Tab. 1 (Dahle et al. , 2017).

The daily average number of tracked GPS satellites was increased from 7.6 to 7.7 after the first update of receiver bandwidth

(van den IJssel et al. , 2016). For LEO satellites, the number of satellites in view is usually larger than 8. But due to the limitation of receiver channels, maximally only 8 GPS satellites can be tracked. There are two reasons for epochs within which less than 7 GPS satellites are tracked.

The first reason is the reacquisition time of the Swarm receiver. When a GPS satellite sets, the receiver cannot immediately track another GPS satellite in view and needs around 84 seconds before the first receiver update (red) and 55 seconds after the

update (blue) to acquire another satellite, shown in Fig. 1 (top), exemplarily for DoY 250, 2015. So, during the reacquisition phase, the number of satellites is less than 8. If some satellites set during short time together, the number of tracked satellites is even less than 7. This shortened reacquisition time is also the main reason for the increased number of tracked GPS satellites.

Another reason for less than 7 tracked GPS satellites is the loss of lock of signal. There are also two reasons for the loss of GPS signal, one is due to the weak signal strength for the GPS satellites at low elevations. However, the loss of lock on GPS

satellites at high elevations occurs mainly at polar and equatorial areas under the influence of ionospheric scintillations, which is evidenced by the change of Total Electron Content (TEC). The occurrence of loss of lock in September 2015 is exemplarily shown in Fig. 1 (bottom). Although the amount of the loss of lock on GPS satellites at higher elevations was decreased from 37 to 11 after the update, the receiver becomes instable for the signals from lower elevations. The amount of the loss of lock increased almost fourfold, from 58 to 256. The reacquisition time in such situation was decreased from around 17 seconds to

11 seconds after the update.



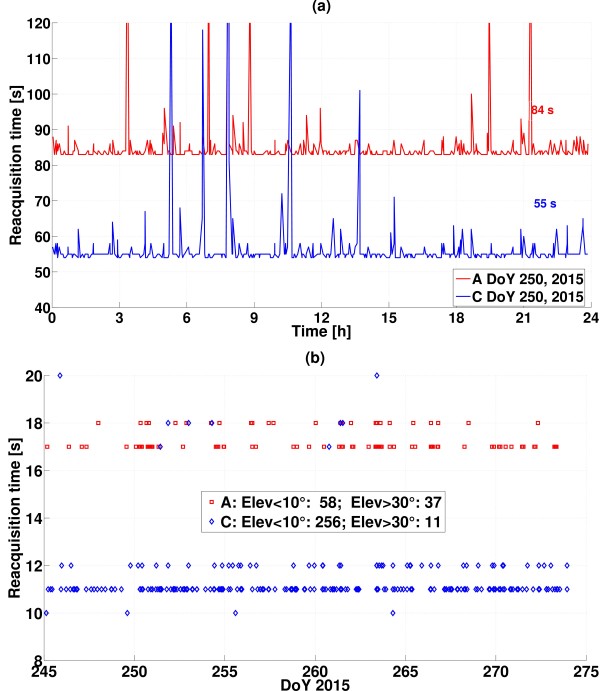

**Figure 1.** Reduction of reacquisition time for Swarm GPS receivers (a) due to setting and rising satellites on DoY 250, 2015 and (b) due to loss of lock in September 2015, red for Swarm A and blue for Swarm C with different tracking settings.

## 2.2 Observation Analysis

As a dual-frequency receiver, the Swarm GPS receiver provides code observations $C/A$ ($C1C$), $P_1$ ($C1P$) and $P_2$ ($C2P$) and carrier phase observations $L_1$ ($L1C$) and $L_2$ ($L2P$) in the Swarm Level 1b product in the RINEX 3.0 format. A sound analysis of the observation noise is mandatory for an adequate stochastic model. The code accuracy is analysed with the multipath linear combination (Estey and Meertens , 1999), which contains multipath effects, ambiguities, code and phase noise. Since carrier phase noise is smaller than code noise by two or three orders of magnitude, it can be neglected. Ambiguities are subtracted as mean value. Then, only code noise and multipath effects remain.

Due to a software issue in the RINEX converter (ESA , 2016), the code observations in the Level 1b product before 11 April 2016 (DoY 102) suffer from high noise. The $P_1$ code noise for all the GPS satellites w.r.t. the elevation for Swarm A on DoY 100, 2016 and DoY 104, 2016 is shown in Fig. 2 as an example. Large code noise of about 2 m ($3\sigma$), even at high elevations, can be seen in the left figure, which is caused by the software issue. After fixing the converter issue, the code noise shows a strong elevation-dependence. The $P_1$ code is disturbed by noise of more than 2 m ($3\sigma$) for satellites at elevations lower than 15°. The noise is smaller than 0.3 m ($3\sigma$) for the satellites above 30°. A $1/sin$ curve fits well with the noise behavior above 15° elevation with a standard deviation ($1\sigma$) of 0.1 m. The standard deviations ($1\sigma$) of $C/A$ code and $P_2$ code for the satellites



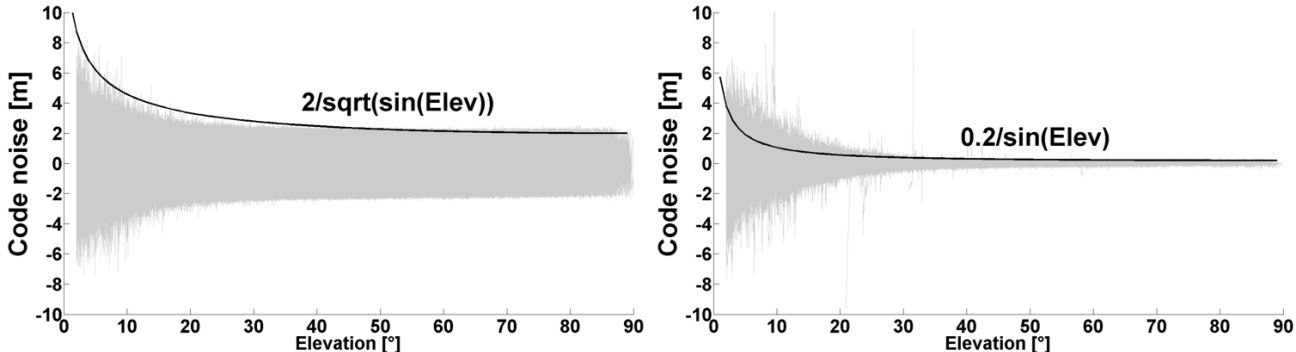

**Figure 2.** $P_1$ Code noise w.r.t. elevation for Swarm A on DoY 100 before fixing the RINEX converter issue (left), and DoY 104, 2016 after fixing the issue (right).

above $15°$ are around 0.33 m and 0.21 m, respectively, after fixing the converter issue. The $C/A$ code noise measured on-board is twice larger than the noise (0.16 m) measured on-ground with double-differences on a zero baseline experiment (i.e. two receivers are connected to one antenna via a signal splitter), reported in Zangerl et al. (2014). The $P_2$ code noise on-board is slightly smaller than in the on-ground test with 0.31 m.

The quality of the kinematic orbit determination depends mainly on the available dual frequency carrier phase observations. To analyse the phase noise, the second differences ($\Delta_2$) of successive phase observations are used, after removing the geometric distances, which are computed with the Swarm Level 2 reduced-dynamic orbits. The deterministic part of the receiver clock errors is removed during differencing. Due to the differencing process involved, this is a qualitative evaluation method rather than a quantitative one.

The resulting $\Delta_2 L_1$ and $\Delta_2 L_2$ phase time series of Swarm A from 18:00 to 24:00 on DoY 333, 2015 for all the GPS satellites are shown in Fig. 3 with respect to time. In addition, the geographic latitude is given. In order to know whether the phase observations are influenced by ionospheric scintillations, the absolute Slant Total Electron Content (STEC), which is defined as integrated electron density along the line of sight from Swarm to GPS satellites and provided in the Swarm level 2 product, with its rate at Swarm A are shown in Fig. 4. The absolute STEC is derived from the geometry-free linear free

combination of $L_1$ and $L_2$, after corrections for differential code biases of GPS satellite transmitters and Swarm receivers (Swarm TEC Product , 2017).

     Figure 3 and 4 highlight the variability of the phase noise with respect to the rate of STEC. Carrier phases on both frequencies are simultaneously disturbed under the influence of ionospheric scintillations for all GPS satellites, even at high elevation. Similar large fluctuations exist also in the ionosphere-free linear combination $\Delta_2 L_3$. This indicates that the fluctuations are

observation noise caused by the GPS receiver and not the variations of ionospheric delays. The large phase noise, even at decimeter level, can be found almost always at high geographic latitude areas (above $60°$, see exemplarily red boxes in Fig. 3),



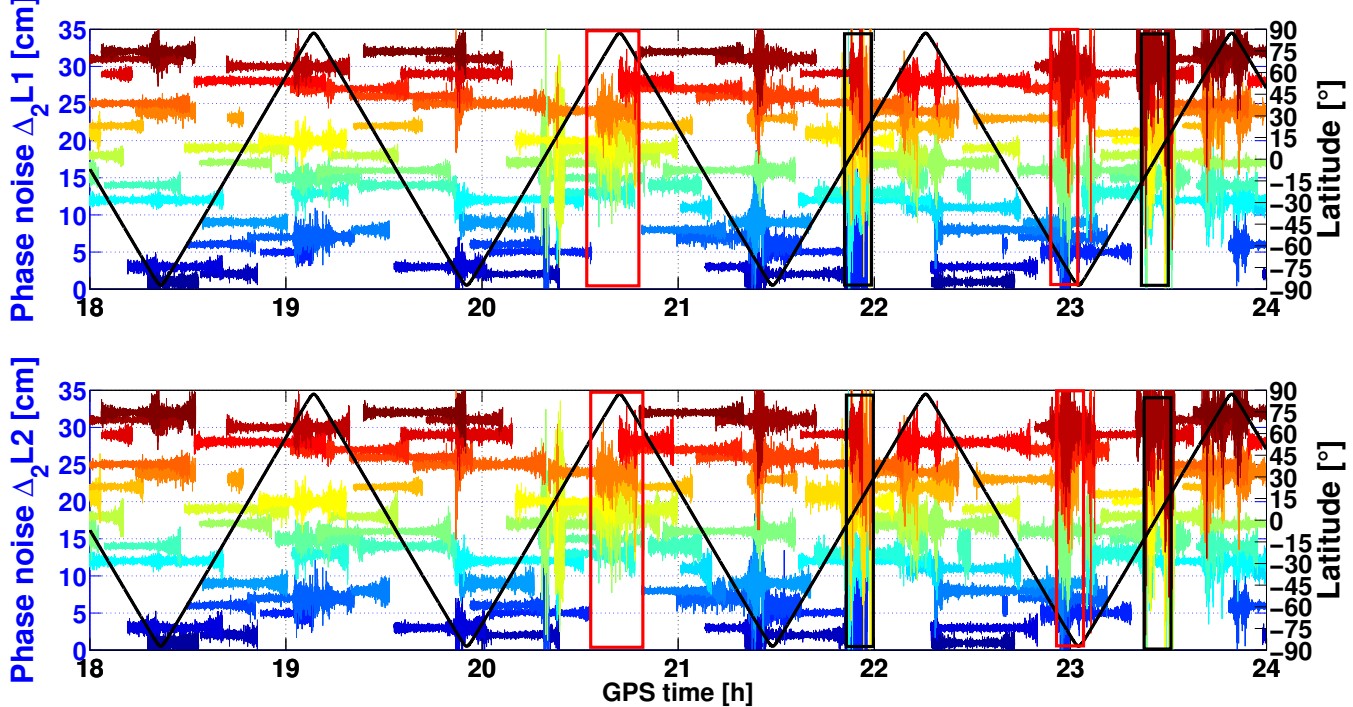

**Figure 3.** $\Delta_2 L_1$ (top) and $\Delta_2 L_2$ (bottom) noise for Swam A from 18:00 to 24:00 on DoY, 333, 2015, color coded per PRN (from blue PRN1 to red PRN 32). The time series are intentionally shifted by PRN × 1 cm. The local time is 8:30 for descending arc and 20:30 for ascending arc. The right legend shows the geographic latitude.

where the STEC is low, but varies rapidly. At mid-latitude areas ($20° - 60°$), the ionosphere is less active, and these effects are rare. At equatorial areas the impact depends on the local time and the activity of local ionosphere. Exemplarily, from 18:00 to 20:00, the STEC is high, but changes slowly which does not disturb the phase observations much. On the contrary, after 20:00 in ascending arcs, with local time around 20:30 after sunset, the changes in STEC are rapid and the phase observations are

5    strongly perturbed (compare black boxes in Fig. 3). This large noise degrade the accuracy of the kinematic orbit significantly and make it difficult to detect small cycle slips with standard methods.

The $\Delta_2 L_1$ and $\Delta_2 L_2$ phase noise at mid-latitude areas w.r.t. elevation is shown in Fig. 5. At a low level of ionospheric activity, the phase noise can reach mm level and the noise on $L_1$ is slightly lower than on $L_2$. The standard deviations ($3\sigma$) of $\Delta_2 L_1$ and $\Delta_2 L_2$ are 8 mm and 9 mm, respectively. There is no strong dependence on the elevation. Assuming independent

10   observations, the standard deviation ($1\sigma$) of the undifferenced phase observations can be assessed to about 1.3 mm for $L_1$ and 1.5 mm for $L_2$ .

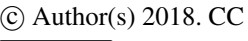

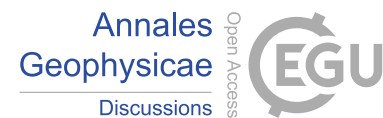

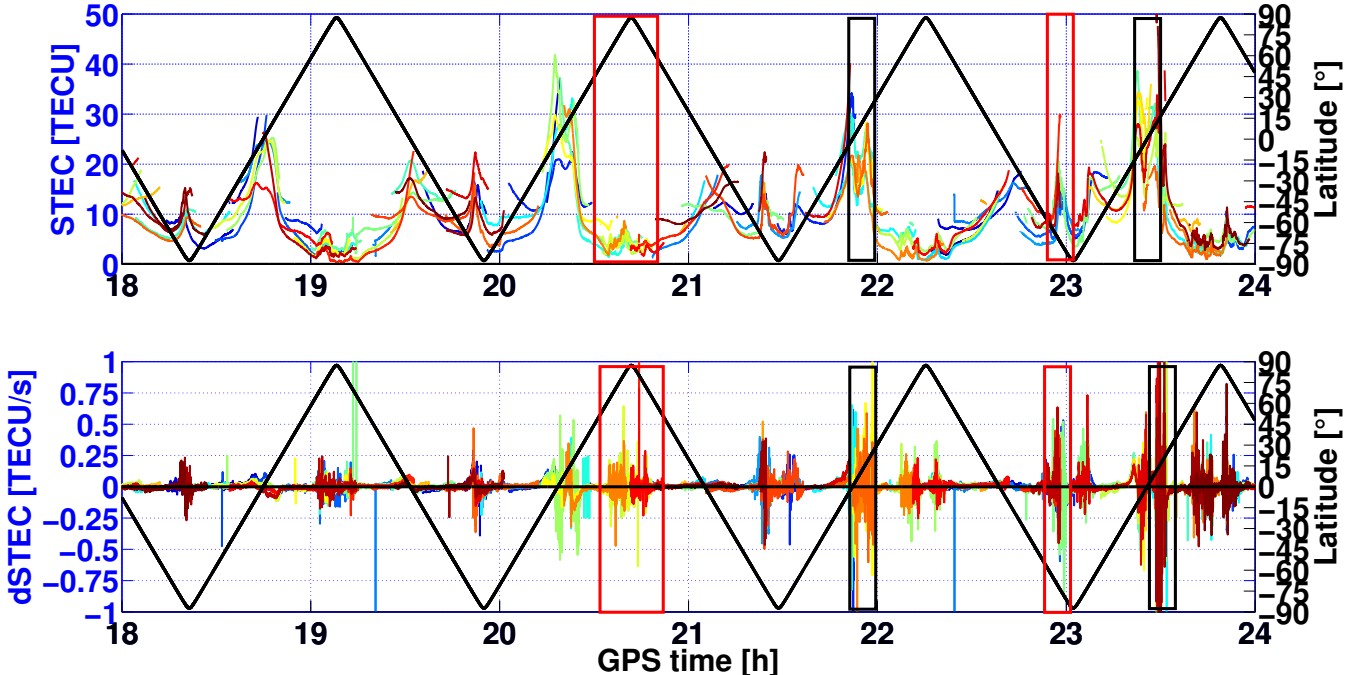

**Figure 4.** STEC (top) and its rate (bottom) for Swam A from 18:00 to 24:00 on DoY, 333, 2015. The right legend shows the geographic latitude.

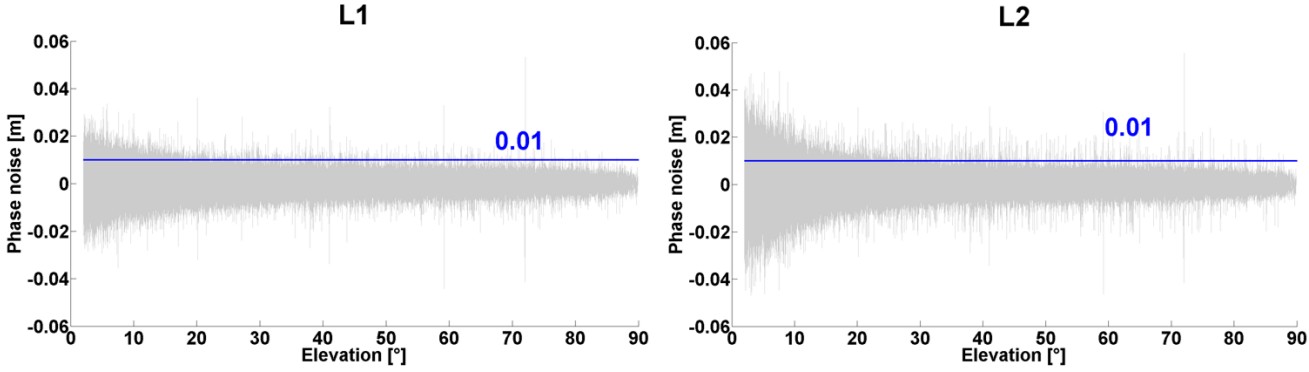

**Figure 5.** Carrier phase noise for $\Delta_2 L_1$ and $\Delta_2 L_2$ at mid-latitude areas on DoY 333, 2015, plotted versus elevation.



### 2.3 Cycle Slip Detect/Repair

Strong ionospheric scintillations cause not only large noise but also cycle slips in the carrier phase observations. Therefore, a proper cycle slips detection is necessary to set up correctly the ambiguities. Since the typical GPS satellite visibility for LEOs is at maximal 40 minutes, the necessity of estimating additional ambiguities for even shorter segments will further decrease the
geometric strength of the positioning.

A classical method for cycle slip detection is to use the TurboEdit method (Blewitt , 1993) based on the Melbourne-Wübbena linear combination (Melbourne , 1985), (Wübbena , 1985):

$$L_{MW} = \frac{f_1 L_1 - f_2 L_2}{f_1 - f_2} - \frac{f_1 P_1 + f_2 P_2}{f_1 + f_2} = \lambda_w (b_1 - b_2),$$ (1)

with

$L_{MW}$     the Melbourne-Wübbena linear combination,

$b_1, b_2$     the ambiguities,

$\lambda_w$     the wide-lane wavelength.

However, the code observations before DoY 102, 2016 contain large noise due to the RINEX converter issue (ESA , 2016), see Fig. 2. The corresponding standard deviation ($1\sigma$) of the Melbourne-Wübbena combination is about 0.8 cycles (wide-lane wavelength). The threshold used in the TurboEdit method for the cycle slip detection is typically $4\sigma$, which means that cycle slips smaller than 3 cycles cannot be detected, see Fig. 6a.

So, a forward and backward moving window averaging (FBMWA) method is applied to reduce the influence of large noise (Cai et al. , 2013). In this algorithm, the widelane ambiguity is smoothed in both forward and backward directions with a specified size m of smoothing window in each direction. This differs from the regular TurboEdit algorithm where only the backward smoothing is performed and the window size continuously grows with the number of epochs. For Swarm satellites, m is set as 50, in order to reduce the noise of widelane ambiguity to 0.1 cycles. As an example PRN23 is shown in Fig. 6b, the
cycle slip is the difference between the forward moving window average at epoch k and the backward moving window average at epoch k-1, here:

$$c_1 = \frac{\Delta L_{MW}}{\lambda_w} = \frac{1}{m \lambda_w} \left( \sum_{i=k}^{k+m-1} L_{MW}(i) - \sum_{i=k-1}^{k-m} L_{MW}(i) \right) = \Delta b_1 - \Delta b_2,$$ (2)

where $\Delta$ is the operator for epoch differencing, and $c_1 = 1$ can be identified in the peak close to an integer (difference smaller than 0.1) in the black curve in this example.

Since with the Melbourne-Wübbena linear combination only the difference of cycle slips on $L_1$ and $L_2$ can be detected and simultaneous occurring of same magnitude cylce slips on $L_1$ and $L_2$ are not detectable, another linear combination is still required, in order to determine the cycle slips on both frequencies, respectively, and repair them. Due to the rapid variations of the ionospheric delays and the large phase noise at ionosphere active areas, the geometry-free linear combination is not suitable





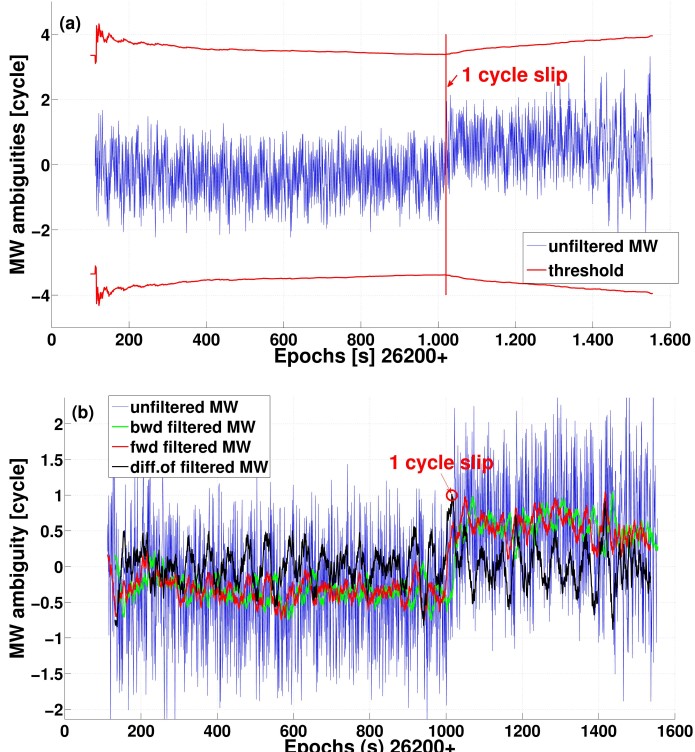

**Figure 6.** Illustration of the proposed cycle slip detection approach based on Melbourne-Wübbena combination, PRN23 of Swarm A on DoY 333, 2015: (a) TurboEdit method, the cycle slip cannot be detected; (b) FBMWA (forward and backward moving window averaging) method $\Delta b_1 - \Delta b_2 = 1$

for Swarm receivers. Here, the time-differenced ionosphere-free linear combination after removing the difference of geometric distances and receiver clock errors is thus used, so that the rapid and large variations of the ionospheric delay can be avoided:

$$\Delta L_3 = \frac{f_1{}^2}{f_1{}^2 - f_2{}^2}\Delta L_1 - \frac{f_2{}^2}{f_1{}^2 - f_2{}^2}\Delta L_2$$
$$= \Delta\rho + \Delta\delta t_r + 0.4844\Delta b_1 - 0.3775\Delta b_2 \tag{3}$$

$L_3$  the ionosphere-free combination of carrier phase
  (other observation errors are corrected),

$\rho$  the geometric distance between GPS satellite and receiver position,

$\delta t_r$  the receiver clock error in meters.





The geometric distances are computed with the Medium Precise Orbit (MEO) provided by ESA in Swarm Level 1b product and removed from the right-hand side of Eq. (3). The prerequisite for this approach is that there is no large jump in the a priori orbit.

After removing the geometric distances term, the differences of receiver clock error and ambiguity on $L_1$ and $L_2$ still remain.
If there is no cycle slip in carrier phase, the ambiguity term should be zero and the difference of receiver clock error can be computed as the average over all tracked GPS satellites, as it is same for all receiver channels. If there are cycle slips for one GPS satellite, the deviation between the average and this satellite is much larger than the other normal satellites. So the satellite with cycle slips can be detected and the difference of receiver clock error should be computed again without this one until no large deviation can be found and removed from the right-hand side of Eq. (3).

After removing the geometric distances and receiver clock error, only the ambiguity term remains. If there is no cycle slip, $\Delta L_3$ time series spreads around zero in accordance with the phase noise. When a cycle slip occurs, a large jump is caused in the time series. In order to determine the cycle slip exactly and distinguish it from outliers, this jump should be computed as accurately as possible. Instead of calculating the epoch difference of two successive epochs (Fig. 7a), the epoch difference of two epochs separated by n epochs (Fig. 7b) are calculated and then the mean value of these m epochs differences is formed
so that the influence of the large noise under ionospheric scintillation can be reduced, see Fig. 7b. Considering the large noise of ionosphere-free linear combination under ionospheric scintillations (around 10 cm), we propose to select n=100, in order to reduce the noise to 1 cm level. Taking the example of PRN23, we can get: $c_2 = 0.4844\Delta b_1 - 0.3775\Delta b_2$, here $c_2 \approx 0.38$.

Together with the equation from Melbourne-Wübbena linear combination, the cycle slip on $L_1$ and $L_2$ can be determined:

$$
\begin{bmatrix} \Delta b_1 \\ \Delta b_2 \end{bmatrix} = \begin{bmatrix} 1 & -1 \\ 0.4844 & -0.3775 \end{bmatrix}^{-1} \begin{bmatrix} c_1 \\ c_2 \end{bmatrix}. \tag{4}
$$

In this example we get: $\Delta b_1 \approx 0.023 \approx 0$ and $\Delta b_2 \approx -1$. If the difference of $\Delta b_1$ to its nearest integer is smaller than 0.2, it can be rounded and marked as "repaired".

The statistics of the repaired cycle slips for Swarm satellites during one year from DoY 245, 2015 to DoY 244, 2016 are listed in Table 2. Around 95% of the cycle slips could be repaired. Issues occur by rapid sequences of cycle slips in short time due to strong ionospheric scintillations. The observations during this period can be excluded as outliers. The majority of the cycle
slips occur on $L_2$. The worldwide distribution of the cycle slips are shown in Fig. 8a. As expected, their occurrence is linked with ionosphere active regions such as polar and equatorial areas. Because of a higher orbit (530 km), where the ionosphere is weaker than at the lower orbit (450 km), the number of cycle slips on Swarm B is less than for Swarm A and C. Due to the similar atmosphere around Swarm A and C, they suffer from the cycle slips almost at the same time. Significantly more cycle slips around the geomagnetic South Pole than around the North Pole are found. The reasons are still under investigation.
In order to see the impact of update of bandwidth of the tracking loop, the cycle slips in September, 2015 are shown in Fig. 8b additionally. It can be seen that the number of cycle slips are significantly reduced after the update in May, 2015 on Swarm C. Because the ionosphere is very weak since May 2016, the tracking is very stable. There were just few cycle slips, so the impact of the update in June and August 2016 cannot be observed.





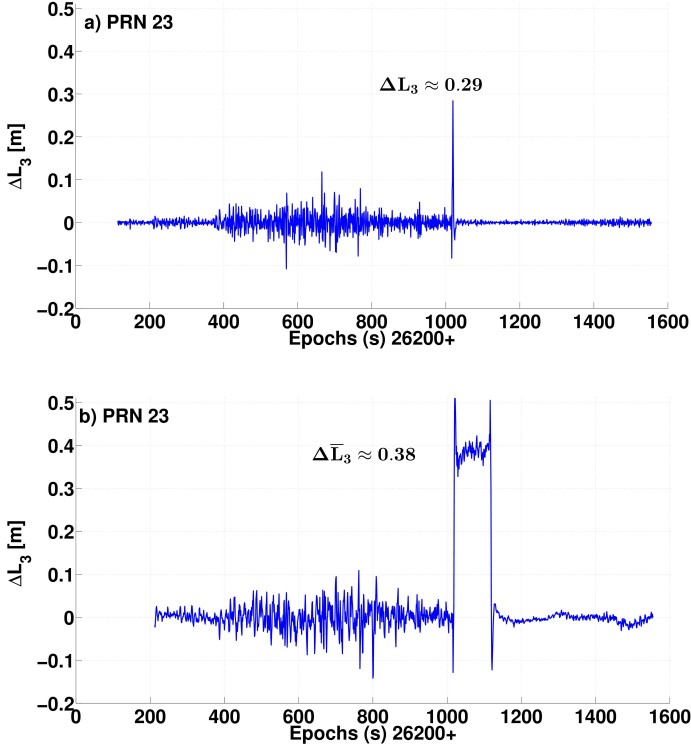

**Figure 7.** Illustration of the proposed cycle slip detection approach based on ionosphere-free linear combination, PRN23 of Swarm A on DoY 333, 2015: a) $\Delta L_3$ between two successive epochs $0.29 = 0.4844\Delta b_1 - 0.3775\Delta b_2$; b) $\Delta L_3$ between two epochs separated by 100 epochs $0.38 = 0.4844\Delta b_1 - 0.3775\Delta b_2$

## 2.4 Outlier detection

As a pure geometric approach, the kinematic orbit is sensitive to the GPS measurement errors. As a consequence, it is important to screen the outliers. The epoch-differenced ionosphere-free linear combination $\Delta L_3$ used in cycle slip detection can also be used to detect the outliers. Considering the L3 noise under ionospheric scintillation can reach 10 cm level, if $\Delta L_3$ is larger than 20 cm, an outlier is detected and should be eliminated. Some systematic errors can also be detected with it. For cases where the orbit of a given GPS satellite is not accurate enough, the time series of $\Delta L_3$ of the corresponding GPS satellite will not be zero-mean, but shows some strong trend. Thus, to avoid a systematic error in the LEO orbit and carrier phase residuals, the corresponding satellites has to be removed accordingly. This can be seen in Fig. 9 for PRN 26 (purple) and PRN 32 (green). Such satellites can be found in the IGS weekly summary (e.g. ftp://ftp.igs.org/pub/product/1864/igs18647.sum.Z). After removing these satellites, the quality of the kinematic orbits are significantly improved. The same effects can also be found on Swarm C at the same time.





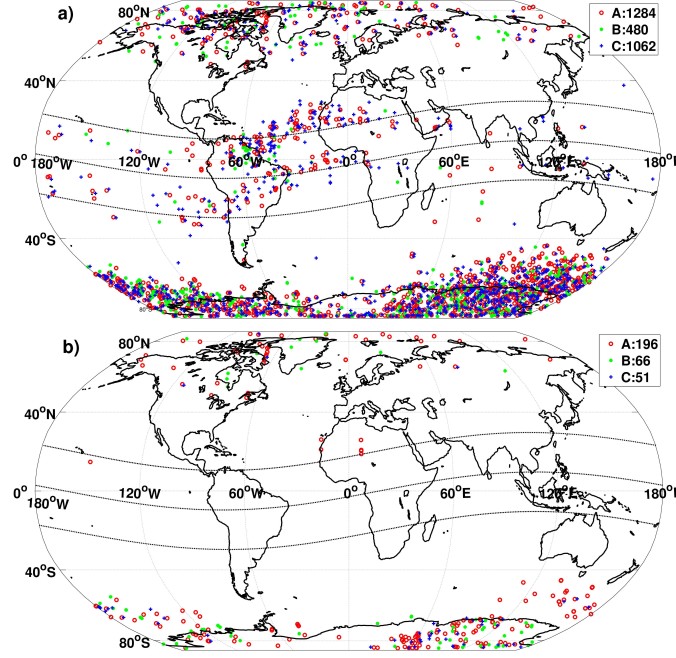

**Figure 8.** Global distribution of cycle slips on Swarm A (red), B (green) and C (blue), a) from DoY 245, 2015 to DoY 244, 2016 and b) for September, 2015. The black lines are the geomagnetic equator and geomagnetic parallel of $\pm 20°$. The results are given in Table 2.

## 3  Kinematic orbit determination

In this section, the approach for kinematic orbit determination at IfE will be introduced first. Then one year kinematic Swarm orbits are validated with the reduced-dynamic orbits and the SLR residuals, respectively. The fully populated variance-covariance matrices of the kinematic orbits obtained from the adjustment are shown and discussed.

5  ### 3.1  Observation modeling

A MATLAB-based PPP software was developed at IfE, Hannover using the least-squares adjustment method. The ionosphere-free linear combinations of P code and phase are computed and introduced as observables in the least-squares adjustment model to eliminate the first-order ionospheric effect. The GPS orbits and satellite clock errors are computed based on the CODE final GPS orbits and 5 s clocks products provided by the Center for Orbit Determination in Europe (CODE) (Dach et al.
10 , 2016)(Bock et al. , 2009). The observation and error modeling used for the kinematic orbit determination are listed in Table 3.

The 1 Hz data is processed over 30 hours, with 3 hours overlap to the previous and following day. It can be avoided that some GPS observations are removed due to the short arc at the day boundary. This increases the stability of the ambiguities at the day boundary. Because the observation time saved in the RINEX file is synchronized to the receiver internal clock, which




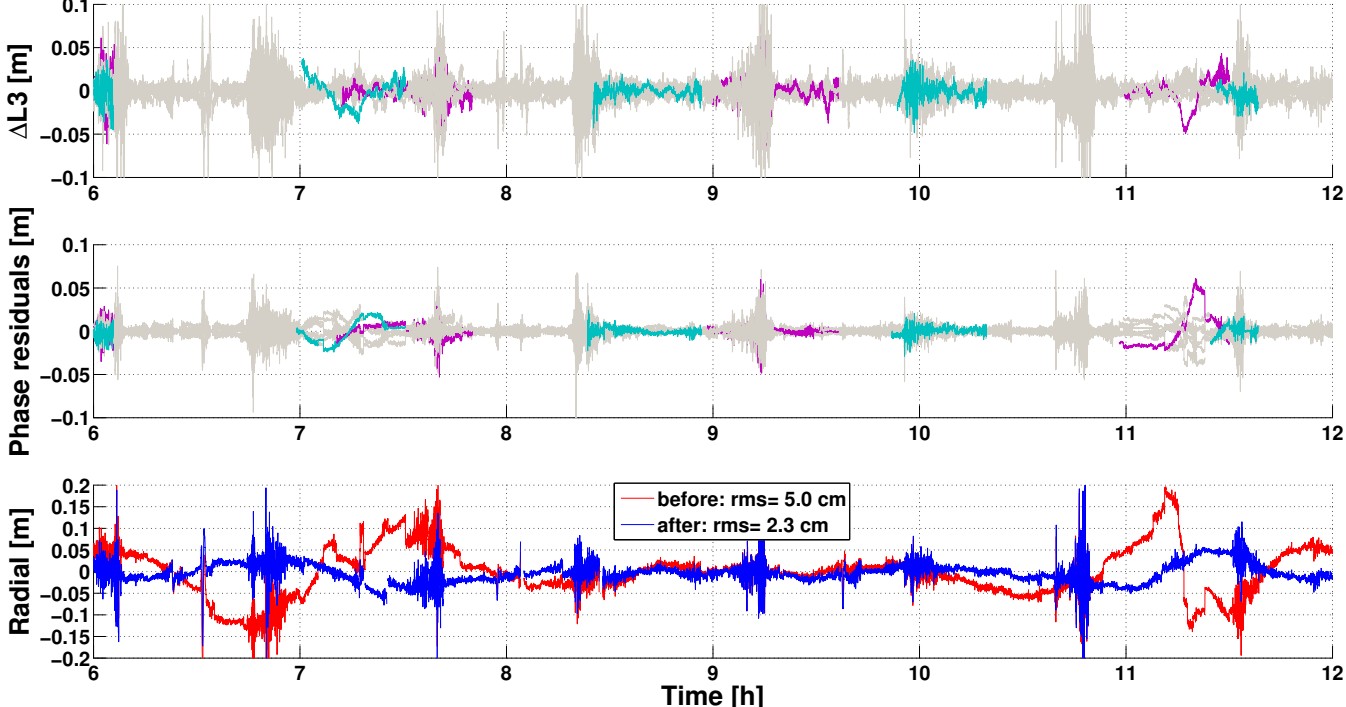

**Figure 9.** Outlier detection and anomalous GPS satellite for Swarm A on DoY 271, 2015, here PRN26 (purple), PRN32 (green). Top panel: $\Delta L_3$ between epochs separated by 100 seconds; middle panel: phase residuals; bottom panel: deviations of the kinematic orbit w.r.t. ESA reduced-dynamic orbits with (red)/without (blue) the anomalous GPS satellites.

has a large clock drift, a time offset is added to synchronize the internal clock with GPS time everyday at the day boundary. This causes a clock jump in the code observations and a clock jump as well as a phase jump in the carrier phase observations. In order to get a continuous arc at the day boundary, these phase jumps need to be fixed. The clock jump is first determined from the average of each epoch-differenced code observations (removing the a priori geometric distances) at the day boundary. Then, the phase jump is the average of each epoch-differenced phase observations (removing the a priori geometric distances) minus the clock jump.

The phase noise is weakly elevation-dependent (see Fig. 5), thus, an identical weight matrix is applied to phase observations. The code noise has a stronger elevation-dependent behavior after fixing the converter issue. So a sine-squared elevation-dependent weight matrix is applied to code observations after fixing, instead of a sine elevation-dependent weight matrix before fixing. The variances are selected with 6 m (before fixing the converter issue)/0.6 m (after fixing the converter issue) for code and 6 mm for carrier phase according to the investigation described in Section 2, taking the error propagation for the ionosphere-free linear combination into account and considering the phase noise under ionospheric scintillation. Accordingly, the weight matrix per epoch has a diagonal structure.



The three unknown position coordinates x, y, z and receiver clock error $\delta t_r$ are estimated epoch-wise, while the ambiguities b are constant over one consecutive satellite arc. A Gauß-Markov model with pre-elimination/back-substitution is used for the estimation of the unknown parameters.

### 3.2 Kinematic orbit results

One year orbits from 02 September 2015 to 31 August 2016 are computed. Here only exemplary but typically results are shown. The position residuals for Swarm A, B and C on DoY 333, 2015 w.r.t. the reduced-dynamic orbits provided by ESA in Swarm Level 2 product are shown in Fig. 10. For a better illustration, the time series for Swarm A and C are shifted by ±10 cm.

The RMS errors in along-track, cross-track and radial directions are around 1.5, 1 and 2 cm, respectively, for all three satellites. The noise caused by the ionospheric scintillations, shown in Fig. 3, degrades the position when passing at equatorial and polar regions. It is mainly absorbed in the radial component. At some ionosphere-active areas the deviations are beyond 10 cm, or even 20 cm. The large noise in the orbits can be eliminated using the Matérn covariance family (Kermarrec et al. , 2018). Comparing Swarm A and C, a similar signature can be seen due to the similar GPS constellation and atmospheric conditions. Besides the large noise, all three orbits are affected by systematic errors in along and radial component, predominantly at timescales to the orbital period. These periodic variations may either be caused by modeling deficiencies in the PPP solution or the reduced-dynamic orbits that are used as reference trajectory.

The RMS errors and average of position residuals for Swarm A, B and C of IfE and ESA kinematic orbits from DoY 245, 2015 to DoY 244, 2016 are listed in Table 4. The days with orbit maneuvers and instrument problems and epochs with GDOP values larger than 5 were excluded. The differences between the reduced-dynamic and kinematic orbits are at the 1.5, 1.5 and 2.5 cm level in the along-track, cross-track and radial directions, respectively. Since Swarm B flies in a higher orbit, the influence of ionosphere is weaker than for Swarm A and C. A better kinematic orbit quality can be observed for Swarm B, especially in the radial direction. It is also worth mentioning that the average offset between kinematic and reduced-dynamic orbit in cross-track is larger than the along-track and radial directions.

The computed kinematic orbits of Swarm C are also compared with the kinematic orbits from ESA (van den IJssel et al. , 2015), IfG of TU Graz (Zehentner and Mayer-Gürr , 2015) and AIUB (Jäggi et al. , 2016). Again, ESA reduced-dynamic orbits are the reference trajectory. The position residuals of all institutes from DoY 01 to DoY 07, 2016 are shown in Fig. 11. The RMS values are given in the legend. Our results are in good agreement with the kinematic orbits derived from the other institutes.

In order to analyse the remaining deviations, a spectral analysis is performed for Swarm C for March 2016 in Fig 12. The once per revolution periodic signatures in the along-track and radial direction are well identified in all orbits except that of IfG, with 93.5 min for IfE and AIUB, 80.3 min for ESA. Besides the once per revolution, a twice per revolution periodic signature can be observed in the three orbits probably due to the impact of ionospheric scintillations at polar areas, with 46.2 min for IfE and ESA, 48 min for AIUB. Interestingly, the signatures of cross track are different; IfE orbits are showing no significant periodicity. A half day periodicity can be observed in AIUB and ESA orbits. The remaining high frequency noise in the radial




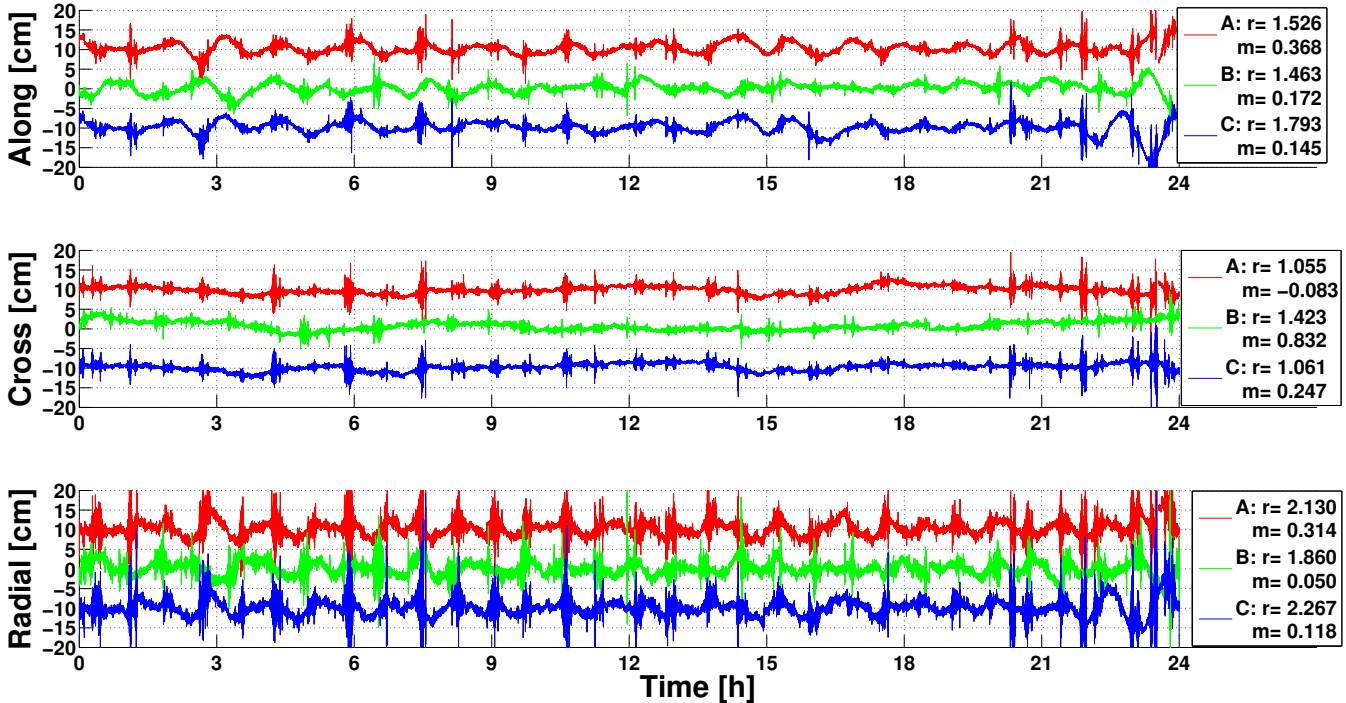

**Figure 10.** Position residuals for Swarm A (red), B (green) and C (blue) w.r.t. ESA reduced-dynamic orbits for DoY 333, 2015. The time series are offset by 10 cm for Swarm A and C. The legend indicates the RMS and mean value in cm.

component is clearly visible in IfE and ESA orbits, which is linked to the observation noise during the passage of ionosphere-active areas. For the along- and cross-track component, IfE is at the AIUB level for frequencies larger than $10^{-3}$ Hz. High frequency white noise contribution dominates in IfE and AIUB orbits until 250 s and ESA and Graz orbits until 100 s. Then flicker noise persist in all the components. Between once per revolution and twice per revolution, in IfE along-track and radial

component some long periodic systematic persist. The cross-track component of IfE is showing a $f^{-1}$ behavior.

An additional possibility to assess the orbit quality is to analyse the residuals of GPS observations (van den IJssel et al. , 2015). The daily RMS errors of ionosphere-free carrier phase residuals for Swarm A, B and C for one year of data from DoY 245, 2015 to 244, 2016 are shown exemplarily in Fig. 13. The average of the daily RMS errors are at the 4 mm level. As expected, the daily RMS errors of Swarm B are slightly smaller than for Swarm A and C. Swarm A and C show similar results.

The RMS value show a decreasing trend with time, which could be related to the decreasing activity of ionosphere. Comparing Swarm A and C in Sept. 2015, the significant improvement due to the update of bandwidth on Swarm C can be observed.

The comparison with reduced-dynamic orbits and the residuals of GPS observations is an internal validation of the orbits. Independent Satellite Laser Ranging (SLR) measurements provide another external validation of the orbits. The GPS-derived kinematic orbits are first interpolated with a polynomial of degree 5 to the observation time of SLR measurements. Then, the

computed ranges between the ground stations of tracking networks and GPS-derived kinematic orbits are compared with the




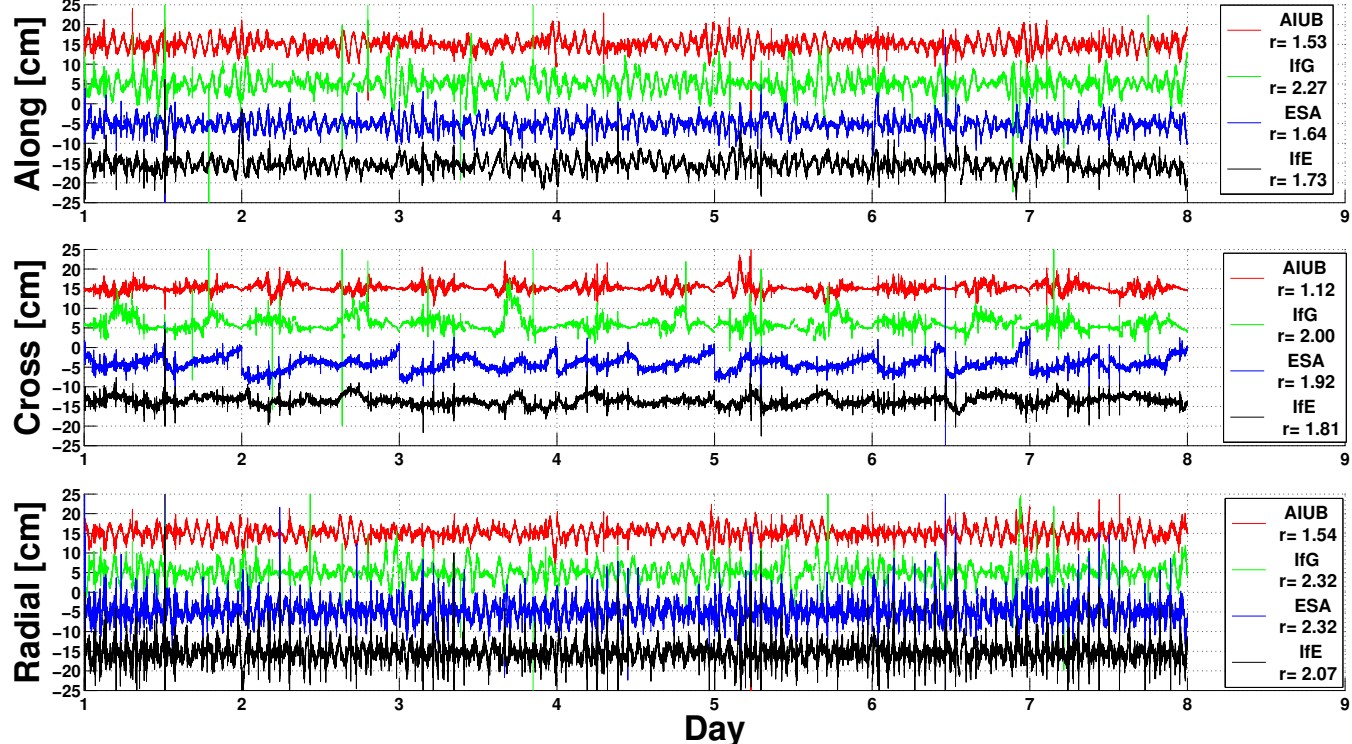

**Figure 11.** Differences of kinematic orbits of Swarm C of different institutes from DoY 1 to 7, 2016. The time series are intentionally shifted by 15, 5, -5 and -15 cm for AIUB (red), IfG (green), ESA (blue) and IfE (black), respectively. The legend gives the RMS in cm.

observed laser ranges. The coordinates of the ground stations are given in ITRF2014 with consideration of solid earth tide and ocean loading (FES2004). The positions of the laser reflector in the satellite reference frame are given in Astrium (1993). The tropospheric delay and relativistic effects are also corrected. The SLR residuals for Swarm A, B and C from DoY 245, 2015 to DoY244, 2016 are shown in Fig. 14. The SLR residuals larger than 30 cm are excluded as outliers. The RMS errors of SLR
5 residuals are at 3-4 cm level for our orbits and ESA kinematic orbits.

### 3.3 Covariance Information

The least-squares estimation provides not only the coordinates but also the variance-covariance information, which is important for gravity field recovery (Jäggi et al. , 2011). Usually, only the mathematical correlations between the coordinates at the same epoch are considered. But the carrier phase ambiguities also correlate coordinates over multiple epochs. The fully populated
10 variance-covariance matrices for 30 hours PPP are computed with 30 s, 10 s and 1 s data rate, respectively. For undifferenced 30 s and 10 s GPS observations, the covariance matrix can be directly computed from the inverse of the normal equation. How-




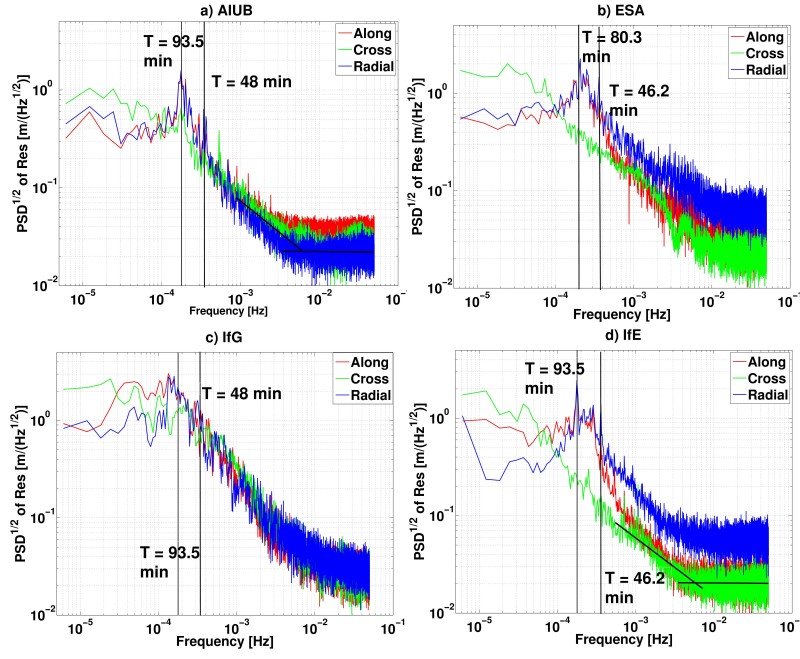

**Figure 12.** Spectral analysis of Swarm C orbits from and AIUB (a), ESA (b), IfG (c) and IfE (d) in March, 2016. Slope 0: white noise; slope -1: flicker noise.

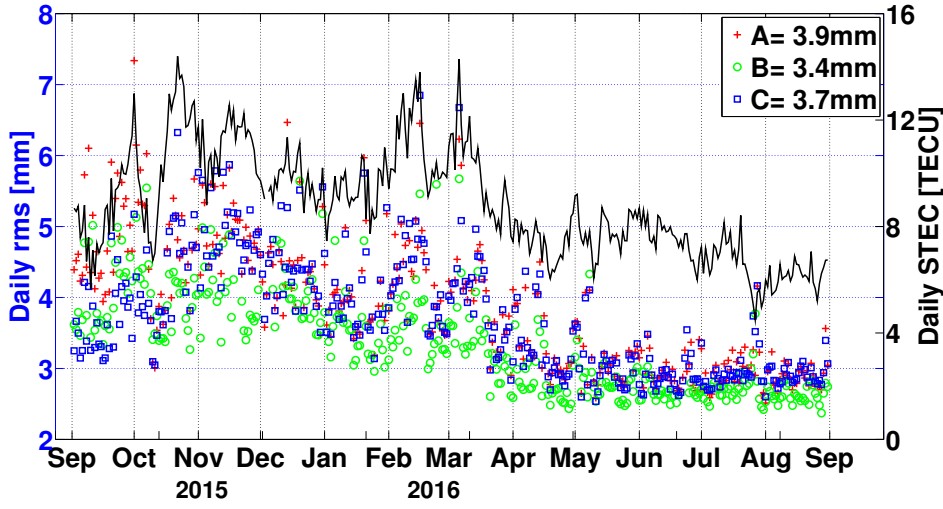

**Figure 13.** Daily RMS errors of ionosphere-free carrier phase residuals and daily average STEC derived from Swarm C over all GPS satellites for one year of data from DoY 245, 2015 to DoY 244, 2016.





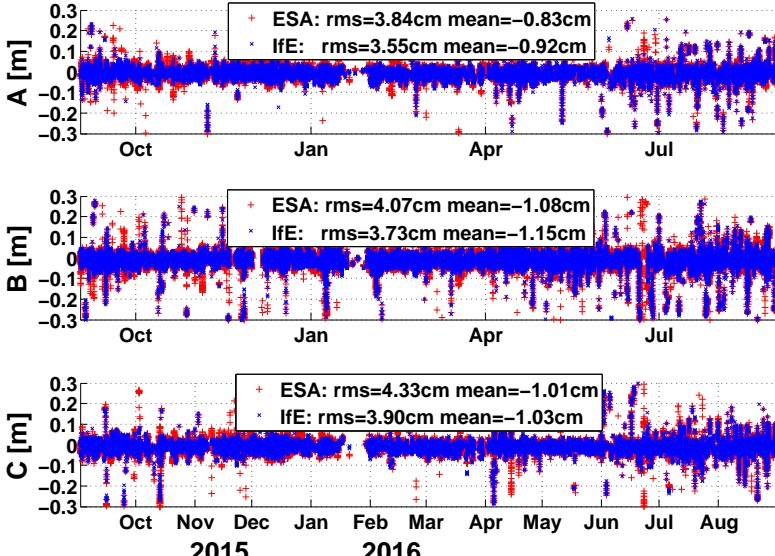

**Figure 14.** SLR residuals over one year from DoY 245, 2015 to 244, 2016, for Swarm A (top), B (middle) and C (botton). The residuals for IfE are given in blue and ESA kinematic orbits in red.

ever, due to computational limitation, the covariance of 1 s is computed epoch-wise. For a better illustration of the covariance matrix, it is converted into a correlation matrix.

Figure 15 shows the corresponding correlation matrix of the Earth-fixed X-component for all 30 hours in the 30 s intervals. The X-coordinates of 30 s rate are strongly correlated at the polar areas (red spots) and the correlation decreases with time (Fig.

15a). The temporal mathematical correlation between two successive 30 s epochs is around 0.6. The correlation time varies between 0.5 and 2.5 hours and it shows a 24-hours variation, which is caused by the period of the GPS constellation. Due to the increased number of observations, the correlation for 10 s observations is much lower than for 30 s. The correlation between two successive epochs is only around 0.4. But it shows a pattern similar to 30 s observations (Fig. 15b). As expected, with increased observations frequency, the correlation for 1 s data rate is further decreased with a correlation between two successive epochs

lower than 0.1 (Fig. 15c). Thus, the increase of data implies an increasing number of coordinates that are less correlated with the ambiguities, since no filter model is used in our PPP approach. The average correlation of X-components with time for the different rates is shown in Fig. 15d. The X-components are still correlated for 30 s data rate after 2 hours. For 10 s data rate, the average correlation is around 0.4 after 10 s and decrease smoothly to 0.1 after 2 hours. There is almost no mathematical correlation for 1 s data rate, the correlation drops immediately to 0.05 after 1 s. The correlation structure of the Earth-fixed

Y-component is similar to the X-component.




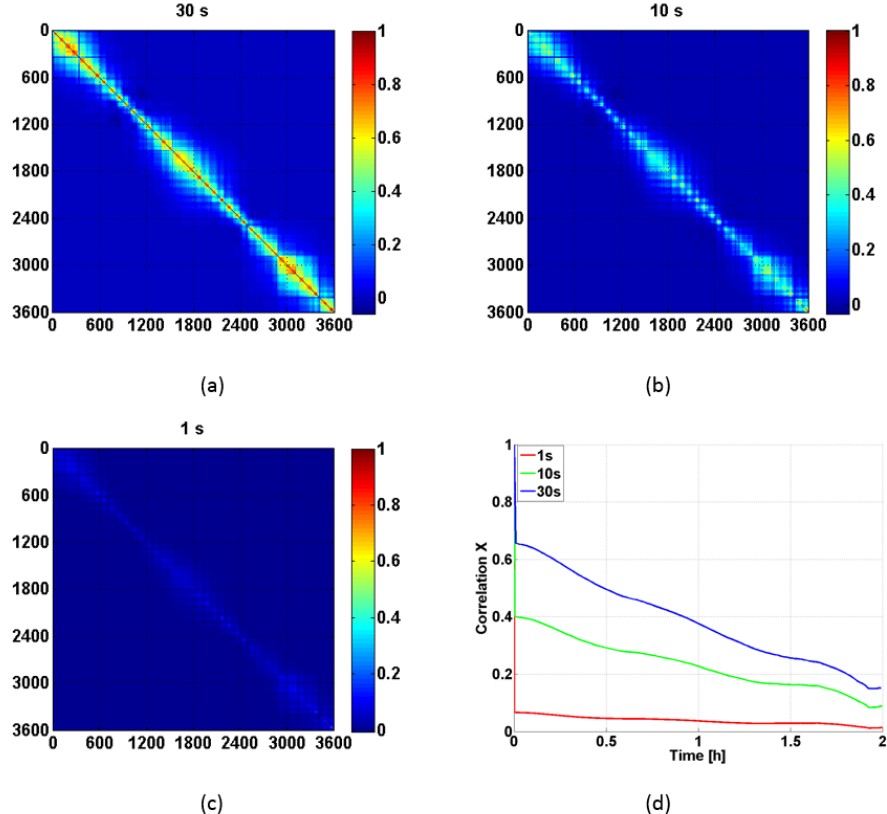

**Figure 15.** Correlation matrix for 30 hours time span in X-component obtained with a 30 s (a), 10 s (b) and 1 s (c) data rate. The identical epochs (30 s interval) are extracted from the respective correlation matrix. Subfigure (d) shows the mean correlation over 2 hours.

The correlation structure of the Z-component for 30 s, 10 s and 1 s data rates are shown in Fig. 16. The correlation between two successive epochs is similar to the X-component, around 0.4, 0.2 and below 0.1 for 30 s, 10 s, and 1 s data rates, respectively. However, the correlation times are much shorter than for the X-component, shown in Fig. 16. After an hour, the correlations can be neglected for 30 s and 10 s data rates.

## 4   Conclusions

In this contribution, we present the approach for determination of the IfE Swarm kinematic orbit with PPP. Since the GPS data quality is a prerequisite for the high quality of kinematic orbit, a thorough analysis was performed first. Although only 8 channels are available, the requirements for an accurate positioning are fulfilled. If the reacquisition time can be shortened, the tracking performance can be further improved. Carrier phase observations at both frequencies are strongly disturbed by ionospheric scintillations, where noise larger than 0.2 m degrades the detection of cycle slips. An innovative approach is




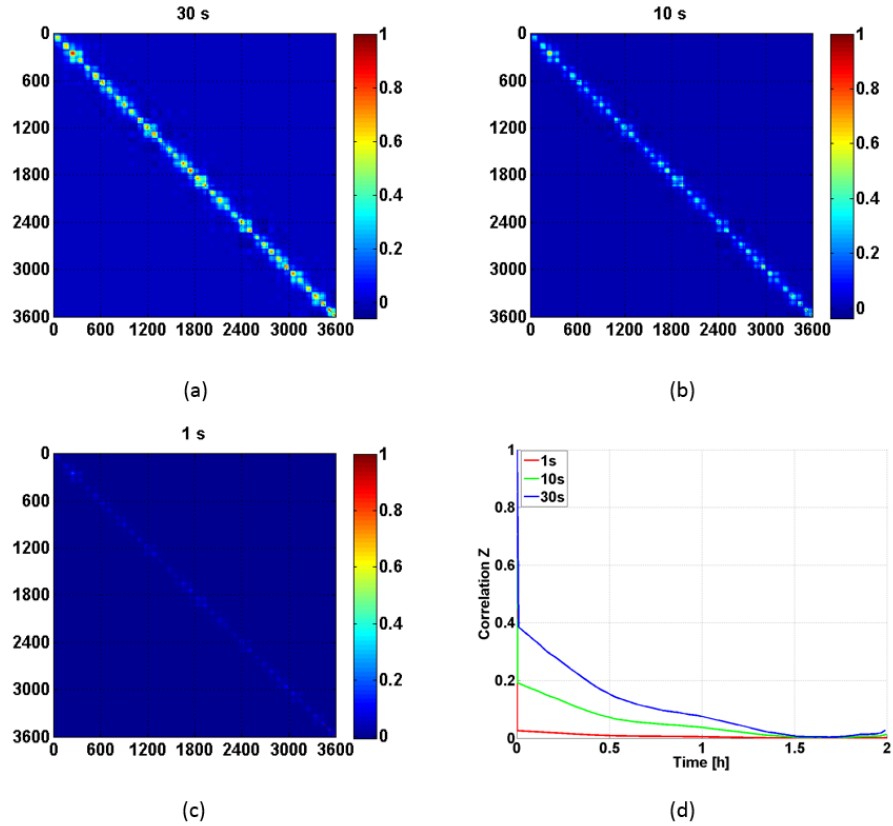

**Figure 16.** Correlation matrix for 30 hours time span in Z-component obtained with a 30 s (a), 10 s (b) and 1 s (c) data rate. The identical epochs (30 s interval) are extracted from the respective correlation matrix. Subfigure (d) shows the mean correlation over 2 hours.

proposed, which allows a successful cycle slips repairing of around $95\%$. It is worth mentioning that almost all the cycle slips occur on $L_2$ due to ionospheric scintillations.

Precise kinematic orbits at IfE are generated with the PPP method. The differences between our kinematic orbits and ESA reduced-dynamic orbits are at 1.5 cm, 1.5 cm and 2.5 cm level in along-track, cross-track and radial directions, respectively.

5   The daily RMS values of ionosphere-free carrier phase residuals are at 4 mm level, showing the good inner consistency. The spectral analysis of the orbits highlights the once per revolution period and higher harmonics as well as the dominant noise processes. A comparison with orbits from other institutes shows very good agreement in RMS level. A comparison with SLR underlines an accuracy of the kinematic orbits of 3-4 cm. Due to the ambiguities in carrier phase, the coordinates are correlated. The mathematical temporal correlation between two successive epochs is larger than 0.6 for 30 s data rate. The

10   correlation decreases with increased sampling frequency. For 1 s data rate, the correlation between two successive epochs is below 0.1.

*Author contributions.* TEXT

*Competing interests.* The authors declare that they have no conflict of interest

*Disclaimer.* TEXT

*Acknowledgements.* This project is part of CONTIM and funded by the Deutsche Forschungsgemeinschaft (DFG) under the SPP1788 Dy-
5 namic Earth which is gratefully acknowledged. We would like to thank ESA for providing Swarm data. The Swarm kinematic orbits have
been made available by ESA, AIUB, and IfG, TU Graz. GPS orbits and clock have been obtained from the Center for Orbit Determination
in Europe (CODE). The SLR residuals are kindly provided by Dr. Anno Löcher of Institute of Geodesy and Geoinformation, University of
Bonn. TU Delft is also very appreciated for providing the PCV maps. The support of all these institutions is gratefully acknowledged.



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



**Table 1.** Update of Swarm carrier loop bandwidth

|  | Swarm A | Swarm B | Swarm C |
|---|---|---|---|
| Before 6 May 2015 | $L_1$: 10 Hz <br> $L_2$: 0.25Hz | $L_1$: 10 Hz <br> $L_2$: 0.25 Hz | $L_1$: 10 Hz <br> $L_2$: 0.25 Hz |
| 6 May 2015 |  |  | $L_1$: 10 Hz->15 Hz <br> $L_2$: 0.25 Hz ->0.5 Hz |
| 8 October 2015 | $L_1$: 10 Hz ->15 Hz <br> $L_2$: 0.25 Hz ->0.5 Hz |  |  |
| 10 October 2015 |  | $L_1$: 10 Hz->15 Hz <br> $L_2$: 0.25 Hz ->0.5 Hz |  |
| 23 June 2016 |  |  | $L_2$: 0.5Hz ->0.75Hz |
| 11 August 2016 | $L_2$: 0.5Hz ->0.75Hz |  | $L_2$: 0.75Hz ->1.0Hz |





**Table 2.** Statistics of the cycle slips for Swarm A, B and C during one year from DoY 245, 2015 to DoY 244, 2016

| Swarm | Total number | repaired number | on $L_1$ | on $L_2$ | repaired percentage |
|-------|-------------|-----------------|----------|----------|---------------------|
| A | 1284 | 1244 | 7 | 1237 | 96.7% |
| B | 480 | 447 | 1 | 446 | 93.1% |
| C | 1062 | 1027 | 4 | 1023 | 96.7% |





**Table 3.** Summary of the measurement and error models used for Swarm kinematic orbit determination

| Model | Description |
| --- | --- |
| GPS tracking data (30 hours) | undifferenced ionosphere-free code and phase |
| GPS Orbits | CODE final GPS orbits and 5s clocks |
| GPS phase model | igs08.atx (week 1888) (Schmid et al. , 2007) |
| Swarm attitude | quaternion from star camera (Level 1b) |
| Swarm phase model | phase center offset (Level 1b) |
|  | phase center variations map (van den IJssel et al. , 2016) |
| stochastic model | $sin(Elev)/(\sigma_c)^2$ or $sin^2(Elev)/(\sigma_c)^2$, $1/(\sigma_p)^2$ |
| a priori coordinates | Medium Accurate Orbit Determination MOD (Level 1b) |
| elevation cut-off angle | $2°$ |
| ionospheric delay | ionosphere-free linear combination |
| phase wind-up | model (Wu et al. , 1993) |
| relativistic corrections | model (IS-GPS-200H , 2013) |
|  | Shapiro effect(Hofmann-Wellenhof et al. , 2008) |





**Table 4.** RMS and mean of position residuals of IfE and ESA kinematic orbits w.r.t. ESA reduced-dynamic orbits for Swarm A, B, C during one year (DoY 245, 2015 to DoY 244, 2016). The values for ESA kinematic orbits are given in brackets.

| Swarm | RMS IfE (ESA) [cm] | | | Mean IfE (ESA)[cm] | | |
|---|---|---|---|---|---|---|
| | Along | Cross | Radial | Along | Cross | Radial |
| A | 1.67 (2.05) | 1.41(1.87) | 2.30 (2.66) | 0.06 (-0.15) | 0.19 (0.30) | 0.12 (0.16) |
| B | 1.47 (1.61) | 1.25 (1.41) | 1.84 (2.12) | 0.05 (-0.14) | 0.27 (0.26) | 0.14 (0.11) |
| C | 1.60 (1.90) | 1.32 (1.61) | 2.28 (2.55) | 0.02 (-0.15) | 0.21 (0.25) | 0.07 (0.14) |