# Peer review of "PPP-based Swarm kinematic orbit determination"

_Annales Geophysicae, 2018_

## Referee Comment (RC1) · Anonymous Referee #1 · 20 Jul 2018

**PPP-based Swarm kinematic orbit determination**

Le Ren[1] and Steffen Schön[1]

[1]Leibniz Universität Hannover, Institut für Erdmessung (IfE), Schneiderberg 50, 30167 Hannover, Germany

**Correspondence:** Le Ren (ren@ife.uni-hannover.de)

It is rather ambitious to state that Swarm can be a gap filler between GRACE and GRACE-FO. It can 
[revised manuscript text omitted]

$L_2$: 0.25Hz | $L_1$: 10 Hz
$L_2$: 0.25 Hz | $L_1$: 10 Hz
$L_2$: 0.25 Hz |
| 6 May 2015 |  |  | $L_1$: 10 Hz->15 Hz
$L_2$: 0.25 Hz ->0.5 Hz |
| 8 October 2015 | $L_1$: 10 Hz ->15 Hz
$L_2$: 0.25 Hz ->0.5 Hz |  |  |
| 10 October 2015 |  | $L_1$: 10 Hz->15 Hz
$L_2$: 0.25 Hz ->0.5 Hz |  |
| 23 June 2016 |  |  | $L_2$: 0.5Hz ->0.75Hz |
| 11 August 2016 | $L_2$: 0.5Hz ->0.75Hz |  | $L_2$: 0.75Hz ->1.0Hz |

[Figure]

**Table 2.** Statistics of the cycle slips for Swarm A, B and C during one year from DoY 245, 2015 to DoY 244, 2016

| Swarm | Total number | repaired number | on $L_1$ | on $L_2$ | repaired percentage |
|---|---|---|---|---|---|
| A | 1284 | 1244 | 7 | 1237 | 96.7% |
| B | 480 | 447 | 1 | 446 | 93.1% |
| C | 1062 | 1027 | 4 | 1023 | 96.7% |

How many cycle slips were not repaired for L1 and how many for L2? Do I overlook?

[Figure]

[Figure]

**Table 3.** Summary of the measurement and error models used for Swarm kinematic orbit determination

| Model | Description |
|---|---|
| GPS tracking data (30 hours) | undifferenced ionosphere-free code and phase |
| GPS  *ephemeris* | CODE final GPS orbits and 5s clocks |
| GPS phase model | igs08.atx (week 1888) (Schmid et al. , 2007) |
| Swarm attitude | quaternion from star camera (Level 1b) |
| Swarm phase model | phase center offset (Level 1b) |
|  | phase center variations map (van den IJssel et al. , 2016) |
| stochastic model | $sin(Elev)/(\sigma_c)^2$ or $sin^2(Elev)/(\sigma_c)^2$, $1/(\sigma_p)^2$ |
| a priori coordinates | Medium Accurate Orbit Determination MOD (Level 1b) |
| elevation cut-off angle | $2°$ |
| ionospheric delay | ionosphere-free linear combination |
| phase wind-up | model (Wu et al. , 1993) |
| relativistic corrections | model (IS-GPS-200H , 2013) |
|  | Shapiro effect(Hofmann-Wellenhof et al. , 2008) |

[Figure]

**Table 4.** RMS and mean of position residuals of IfE and ESA kinematic orbits w.r.t. ESA reduced-dynamic orbits for Swarm A, B, C during one year (DoY 245, 2015 to DoY 244, 2016). The values for ESA kinematic orbits are given in brackets.

| Swarm | RMS IfE (ESA) [cm] | | | Mean IfE (ESA)[cm] | | |
|---|---|---|---|---|---|---|
| | Along | Cross | Radial | Along | Cross | Radial |
| A | 1.67 (2.05) | 1.41(1.87) | 2.30 (2.66) | 0.06 (-0.15) | 0.19 (0.30) | 0.12 (0.16) |
| B | 1.47 (1.61) | 1.25 (1.41) | 1.84 (2.12) | 0.05 (-0.14) | 0.27 (0.26) | 0.14 (0.11) |
| C | 1.60 (1.90) | 1.32 (1.61) | 2.28 (2.55) | 0.02 (-0.15) | 0.21 (0.25) | 0.07 (0.14) |

Dear authors,

It was a pleasure to read your manuscript. It is my opinion it can be accepted for publication after a minor revision. Please observe my annotations, I think they speak for themselves and can be easily handled.

Kind regards,
Reviewer

---

## Referee Comment (RC2) · Anonymous Referee #2 · 30 Jul 2018

Dear author, In your paper "PPP-based Swarm kinematic orbit determination" you discuss an alternative processing strategy for Swarm kinematic orbits. First you analyze the tracking performance, then you describe your outlier, cycle slip detection schema and correction for the day boundary. At the end you compare your results to other providers and analyze the covariance matrix. First of all I want to thank you for the nice presentation of your work the text is well written and the steps you took in the processing are well motivated and can be understood.

My major concerns belong to chapter 2.2:

1. First paragraph: sound analysis means frequency analysis? Did you do this? I only see time dependent or elevation dependent plots.

[Figure]

2. Second paragraph: 1/sqrt(sin) you should mention where this dependence is coming from and what could be the cause why it is not fulfilled after fixing the RINEX converter. I would not even state that it fits well before the fixing. Please discuss in more detail what a deviation from 1/sqrt(sin) means.

3.Third paragraph: second difference is not a good expression for the difference of differences, please explain in more detail.

4.With the construction of these differences of the differences a problem is accompanied, namely in giving this quantity an unit. You chose m (as it is a difference of two meter values) but this value is in some parts dependent on the sampling rate so $m/s^2$? but this is also not really correct as we do not talk about accelerations. You can stick to meter, but you should be aware that is this somehow an arbitrary unit and it is only useful in comparing the same data set. What you mention in the text. You should make this clear and I would even prefer to indicate the fact in the plots by using "arbitrary units". In this sense also your paragraph 6 where you discuss the 8 and 9mm noise level of this quantity is not adequate. You should just compare xx times higher in the regions of the poles and equator.

5. You should also stick to standard deviation 1sigma or 3sigmas but do not mix it. 6. Paragraph 4 is wrong: difference of differences of L3 can have strong influence left from the ionospheric fluctuations. Two frequencies do not take exactly the same path through the ionosphere and therefore depending on the size of the fluctuations can have a totally different instantaneous effect on L1 and L2 what you extract by building the differences. So this is probably no issue of the receiver.

Section 2.3

1. Equation 3 b1 and b2 have no unit (but in the equation meters are needed)

2. Line 14 page10: what does n stand for? Which value does it have?

Minor points:

1. Page 2 line 26 An approach . . .

2. Page3 paragraph 2 and 3 belong together

3. Page 3 line 23: Another reason for tracking less than . . .

4. Page 5 line 8: errors

5.Page 6 line 5 degrades

Kind regards, the reviewer

---

## Author Comment (AC1) · 31 Aug 2018

Dear Reviewer,

Thank you very much for your reviewing and so many valuable comments. I am very glad to answer your questions one by one.

Our answers are summrized in the attached pdf file.

Best regards,

Le Ren and Steffen Schön
* * *
—— PS: I am not very sure, if you can open the attached pdf file, so I put my answers

also here:

Dear Reviewer,

Thank you very much for your reviewing and so many valuable comments. I am very glad to answer your questions one by one.

P.1. L.1 : It is rather ambitious to state that Swarm can be a gap filler between GRACE and GRACE-FO. It can provide temporal gravity field information for the gap, but not with the same quality and spatial resolution.

A: We changed the sentence according to your suggestions.

P.1. L.14: Please specify for which purpose(s) the correlation needs to be taken into account.

A: I added " for example, for the recovery of gravity field from kinematic orbits" after "taken into account"

P.2. L.5: A: the -> independent

P.2. L21: Please note that for the ESA orbit products use is made of the DLR/GSOC GHOST software. Thus this is not an ESA estimator. Please correct.

A: changed to "Bayesian weighted least-squares estimator, which is implemented in the GPS High-precision Orbit Determination Software Tools (GHOST) developed at the Deutsches Zentrum für Luft- und Raumfahrt in close cooperation with TU Delft and used for the ESA official orbits".

P.3 L.1 : It also relies on the quality of GPS ephemeris and clock solutions!

A: added "as well as on the introduced GPS orbit and clock products".

P.3. L.10

A: 10° -> 10° elevation

P3. L.28 and L.29: Do these numbers apply for daily period?

A: No, they are for monthly period, in September 2015, which can be seen in Fig.1b.

P.5. L.4: 21 cm is not slightly smaller than 31 cm?

A: we removed "slightly".

P.5 L.6: Exactly how is "second differences" defined? Is the difference between two consecutive observations at a time step of one second?

A: not exactly, the second differences is the difference of differences between two consecutive observations at a time step of one second. We changed "second differences" to "second-order differences" , which is often used in the time series analysis and added a formula $\Delta_2 L(t)=L(t+1)-2L(t)+L(t-1)$ for a better understanding.

P.5 L.20: Are you sure, how about higher order ionospheric effects? Ionospheric scintillations? Am I missing the point here? Please explain.

A: The higher order ionospheric effects cannot reach decimeter level. According to the study of Jäggi (2015), the effects of higher order ionospheric effects can be neglected.

It is difficult to assess the actual reason of the noise, thus, we changed the sentence.

We delete the sentence "This indicates that the fluctuations are observation noise caused by the GPS receiver and not the variations of ionospheric delays."

Actually, at least some of these errors are caused by the GPS receivers, which are sensitive to differential dynamic, e.g. due to ionosphere (private communication with Franz Zangerl). After the update of carrier loop bandwidth (L1: from 10 Hz to 15 Hz, L2: from 0.25 Hz to 0.5Hz), these systematic errors are significantly reduced.

P.6 L.5 & 6

A: degrade -> degrades; make -> makes

P.6 L.8: standard deviation is 1-sigma, not 3-sigma. Please be consistent with terminology and symbols. This is confusing ...

A: sorry for the confusion, I have changed all of them to 1 sigma, also for the code noise part. Because the 3-sigma are directly estimated from the figure and the 1-sigma is computed from the data, so, the computed 1-sigma is not exactly 1/3 of the 3-sigma value used in the original text.

P.8 L.1:

A: Detect-> Detection

P.10 eqa. (1): Please check if all symbols have been described (before).

A: Thank you for the remind, I added f1 f2 L1 L2.

P.10 L.17 & 19: m in Italics? Now, it suggests "meter". Also, what does "m" represent, is it number of observations?

A: sorry for the confusion. Here m should be in italics, m is the length of moving-average window, namely, the number of observations.

P.10 L.5: zero constant?

A: No, it is not a constant zero, but zero-mean, because of noise.

P.10 L.14 n?

A: sorry for it, here m should be n.

P.11 L.5 : This is only two times 10 cm. What is the confidence interval?

A: here 20 cm is just an empirical value, derived from the time series in the figure, there is no strict threshold for it. Theoretically, the confidence interval should be 95%.

P.12 L.6: Reference or copyright sign?

A: This is developed by ourselves.

P.14 L.2 :

A: satellite arc -> GPS satellite-to-satellite tracking pass.

P.14 L.2: What are the parameter settings for the Gauss-Markov process, e.g. correlation time, a priori and steady-state sigmas? Did I overlook?

A: These information are listed in Table 3. A batch lease-squares adjustment is used, so no noise processes are set up.

P.14.L.12: The large noise in the orbits can be eliminated using the Matérn covariance family (Kermarrec et al. , 2018). Did you use this? If yes, then please describe how, otherwise remove this remark.

A: We do not use this method to the results in this paper. We deleted it.

P.14. L.31: Can you explain the significant difference between these two values, i.e. 93.5 and 80.3 min?

A: We are not very sure about the reason for the 80.3 min for ESA. These PSD are derived from the computed orbits, so it is normal that there is a difference between the real and theoretic value. Some uncorrected systematic errors can also change the period of the time series, for example, undetected cycle slips.

P.15 Fig.10

A: added "RMS = r, mean =m".

P.15 L.3: Do you mean periods of 250 s and 100 s?

A: Yes.

P.16 Fig.11

A: added "with respect to ESA reduced-dynamic orbits".

P.16 L.1: ITRF2014 FES2004 Please include references for ITRF2014 and FES2004.

A: added Altamimi(2016) for ITRF2014 and Lyard et al. (2006) for FES2004

P.16 L.5: Did you also apply elevation and azimuth corrections for the Swarm SLR reflectors?

A: No, reflector-dependent range corrections were not applied. This information is added to the text. P.18 L.1: the covariance of 1 s is computed epoch-wise, Please describe how?

A: We added "the covariance of 1~s is computed epoch-wise by sequential inversion of block submatrices", the formulas are listed below but not shown in the paper.

P.18 L.6: 24 hr is two times the GPS orbital period?

A: Yes.

P.20 L.5

A: inner -> internal

P.21 L.3:

A: clock->clocks

P22. L.36: No authors for this technical report?

A: No, this is an offical document from ESA. We do not find the authors' name.

P.25 Table2: How many cycle slips were not repaired for L1 and how many for L2? Do I overlook?

A: The number of unrepaired cycle slips is same for L1 and L2. The cycle slips on L1 and L2 are repaired together, which means, if we can repair L1, L2 can also be repaired, if we cannot repair L1, L2 cannot be repaired neither.

P.26 Table:

A: Orbits-> ephemeris

Please also note the supplement to this comment:
https://www.ann-geophys-discuss.net/angeo-2018-52/angeo-2018-52-AC1-supplement.pdf
* * *

---

## Author Comment (AC2) · 31 Aug 2018

Dear Reviewer,

Thank you very much for your reviewing and so many valuable comments. I am very glad to answer your questions one by one.

Chapter 2.2:

1. First paragraph: sound analysis means frequency analysis? Did you do this? I only see time dependent or elevation dependent plots.

A: I'm sorry, this is a small misunderstanding, here "sound" means "solid", not the spectral analysis. I have changed it, used "solid" instead of "sound".

2. Second paragraph: 1/sqrt(sin) you should mention where this dependence is coming from and what could be the cause why it is not fulfilled after fixing the RINEX converter. I would not even state that it fits well before the fixing. Please discuss in more detail what a deviation from 1/sqrt(sin) means.

A: We added a sentence on typical observation weighting in GNSS processing; $1/\sqrt{\sin(Elev)}$ is one of the models often used to assess weak elevation dependence compared to $1/\sin(Elev)$.

This weaker dependence is due to a software issue in the RINEX converter and reported in the ESA website, but there is no detailed information about the reason and effects of this issue. ( https://earth.esa.int/web/guest/missions/esa-operational-eo-missions/swarm/news/-/asset_publisher/K3vp2LwLXSrF/content/swarm-software-issue-in-rinex-converter-fixed?redirect=https%3A%2F%2Fearth.esa.int%2Fweb%2Fguest%2Fmissions%2Fesa-operational-eo-missions%2Fswarm%2Fnews%3Fp_p_id%3D101_INSTANCE_K3vp2LwLXSrF%26p_p_lifecycle%3D0%26p_p_state%3Dnormal%26p_p_mode%3Dview%26p_p_col_id%3Dcolumn-1%26p_p_col_pos%3D1%26p_p_col_count%3D2%26_101_INSTANCE_K3vp2LwLXSrF_cur%3D12%26_101_INSTANCE_K3vp2LwLXSrF_keywords%3D%26_101_INSTANCE_K3vp2LwLXSrF_advancedSearch%3Dfalse%26_101_INSTANCE_K3vp2LwLXSrF_delta%3D10%26_101_INSTANCE_K3vp2LwLXSrF_andOperator%3Dtrue ).

The elevation dependence of the noise in GNSS is under nominal reception conditions- covered by the antenna gain pattern. For the spacecraft an exponentially elevation dependent pattern could fit even better the residual time series. A deviation from the curve shows that the precision of the observations at that specific elevation is either overestimated or underestimated.

3. Third paragraph: second difference is not a good expression for the difference of differences, please explain in more detail.

A: Thanks for this comment. .We changed "second differences" to "second-order differences" , which is often used in the time series analysis and added a formula $\Delta_2 L(t) = L(t+1) - 2L(t) + L(t-1)$ for a better understanding.

4. With the construction of these differences of the differences a problem is accompanied, namely in giving this quantity an unit. You chose m (as it is a difference of two meter values) but this value is in some parts dependent on the sampling rate so m/s2? but this is also not really correct as we do not talk about accelerations. You can stick to meter, but you should be aware that is this somehow an arbitrary unit and it is only useful in comparing the same data set. What you mention in the text. You should make this clear and I would even prefer to indicate the fact in the plots by using "arbitrary units". In this sense also your paragraph 6 where you discuss the 8 and 9mm noise level of this quantity is not adequate. You should just compare xx times higher in the regions of the poles and equator.

A: With the added definition of "second differences" and the formula, it is more clear that the unit is meter.

The noise for L1 and L2 are computed from $\Delta_2$L1 and $\Delta_2$L2 again, according to the error propagation law, $\sigma_{L1} = \sigma_{\Delta_2 L1}/\sqrt{6}$, same for L2.

5. You should also stick to standard deviation 1sigma or 3sigmas but do not mix it.

A: Sorry for the confusing with 1/3 sigma. We have changed them to 1 sigma.

6. Paragraph 4 is wrong: difference of differences of L3 can have strong influence left from the ionospheric fluctuations. Two frequencies do not take exactly the same path through the ionosphere and therefore depending on the size of the fluctuations can have a totally different instantaneous effect on L1 and L2 what you extract by building the differences. So this is probably no issue of the receiver.

A: We agree that it is difficult to assess the actual reason of the noise, thus, we changed the sentence.

 We delete the sentence "This indicates that the fluctuations are observation noise caused by the GPS receiver and not the variations of ionospheric delays."

Actually, at least some of these errors are caused by the GPS receivers, which are sensitive to differential dynamic, e.g. due to ionosphere (private communication with Franz Zangerl). After the update of carrier loop bandwidth (L1: from 10 Hz to 15 Hz, L2: from 0.25 Hz to 0.5Hz), these systematic errors are significantly reduced.

Section 2.3

1. Equation 3 b1 and b2 have no unit (but in the equation meters are needed)

A: Yes, b1 and b2 have no unit, the unit is contained in 0.4844 and 0.3775. We added formula instead of these values, so, the unit problem can be avoided.

2. Line 14 page10: what does n stand for? Which value does it have?

A: n is the number between two epochs used in the computation of $\Delta L_3$, that means $\Delta L_3(t) = L_3(t+n) - L_3(t)$. Considering the noise level, we propose n=100 for Swarm satellites.

Minor points:

1. Page 2 line 26 An approach . . .

A: corrected

2. Page3 paragraph 2 and 3 belong together

A: I think, it is too long to put paragraph 2 and 3 together.

3. Page 3 line 23: Another reason for tracking less than . . .

A: corrected

4. Page 5 line 8: errors

A: corrected

5.Page 6 line 5 degrades A: corrected

---

## Author Comment (AC3) · 31 Aug 2018

Dear Reviewer,

I also upload the revised paper in the attachment.

Best regards,

Le Ren

Please also note the supplement to this comment:
https://www.ann-geophys-discuss.net/angeo-2018-52/angeo-2018-52-AC3-supplement.pdf